# Sustainable Transformation of Resettled Communities for Landless Peasants: Generation Logic of Spatial Conflicts

**Kexi Xu** [1,2,3], **Hui Gao** [2], **Haijun Bao** [1,*], **Fan Zhou** [2] and **Jieyu Su** [2]

1   School of Spatial Planning and Design, Zhejiang University City College, Hangzhou 310015, China; xkxzj2017@zufe.edu.cn
2   School of Public Administration, Zhejiang University of Finance and Economics, Hangzhou 310018, China; gaohui@zufe.edu.cn (H.G.); zfzj2019@zufe.edu.cn (F.Z.); sujieyu@zufe.edu.cn (J.S.)
3   School of Public Administration, Zhejiang University, Hangzhou 310030, China
*   Correspondence: baohaijun@zucc.edu.cn

**Abstract:** Urbanization in China has resulted in serious conflicts. Landless peasants are resettled between urban and rural areas in transitional communities. where their rural lifestyles often lead to spatial conflicts. We proposed a conceptual model to provide theoretical guidance for the governance of spatial conflicts and the sustainable transformation of resettled communities. Using field observations and semi-structured interviews, we examined 10 resettled communities in Hangzhou, China. The use of grounded theory to code the interview texts yielded 71 initial concepts and 22 categories that we then refined into six main categories: community physical environment (e.g., quality of private housing), community communication environment (e.g., heterogeneity of community population), landless peasants' risk perceptions (e.g., impacts on social psychology), community governance capacity (e.g., trust in community's self-governing organizations), residents' space perceptions (e.g., awareness of space rights), and space competition behavior (e.g., fighting for public space). Finally, we applied social combustion theory to construct a logical relationship between the core category and main categories. The results show that changes in the physical and communication environments are the root elements ("combustion substances") of spatial conflicts; the driving factors are landless peasants' risk perceptions and community governance capabilities; direct elements ("ignition temperature") are residents' space perceptions and space competition behavior. Strategies for sustained transformation in resettled communities should prioritize gradual transitions of community space, improve support mechanisms for landless peasants, optimize community governance mechanisms, and cultivate awareness of community rules. This study aids the understanding of the inner mechanism for the sustainable development of resettled communities and has implications for other countries and regions in similar contexts.

**Keywords:** community spatial conflict; landless peasants; the resettled community; community governance; social combustion theory; grounded theory

## 1. Introduction

In recent decades, China has been experiencing urbanization on an unparalleled scale in human history. Statistics from the National Bureau of Statistics of the People's Republic of China show that the urbanization rate in China has increased from 17.9% in 1978 to 63.89% in 2021; the rate is expected to reach 80% by 2050. China will finish the urbanization process over several decades, while western countries have completed it over several hundred years. Therefore, the process is referred to as spatio-temporal compressive urbanization [1]. The urbanization process in China is promoted by the expropriation of entire villages, including their farmlands and housing plots, for conversion to urban land [2]. Therefore, numerous landless peasants migrate from traditional villages to resettled communities.

The resettled community for landless peasants is a transitional community between urban and rural communities. Behaviors associated with rural life habits, such as occupying

public space as if it were private, competing for parking spaces, holding weddings and funeral ceremonies in open spaces, growing vegetables by destroying green belts, burning ghost money in the corridor, and creating noise pollution [3–5], are common in resettled communities. These behaviors of space competition have caused a large number of conflicts between community actors [6]. In addition, the reshaping of social relations in resettlement communities has not kept pace with the influx of new immigrants, further aggravating problems related to spatial conflicts. Consequently, the accumulation of such conflicts leads to the involution of landless farmers, which affects the outcome of urbanization. Moreover, without timely resolution of such conflicts, the resettled communities for landless farmers will become breeding grounds for "slums" and "shanty towns" and create considerable challenges to the sustainable development of cities [7,8].

The resettled community for landless peasants is a product unique to China's urbanization process. However, its problems are similar to those faced by resettled communities formed by involuntary resettlement projects in other countries or regions worldwide. Owing to natural disasters, wars, slum clearance plans, and large-scale infrastructure projects, approximately 10 million people worldwide are resettled to new communities every year [9–12]. The initial goal of the involuntary resettlement project is to improve living environments. However, there are often intense community conflicts within immigrant groups, between different immigrant groups, and between immigrant groups and the original residents of such communities, that affect their sustainable development [13,14]. Among these, conflicts caused by space competition behaviors are pervasive in resettled communities. It mainly manifests in housing space conflicts and community public space conflicts. For example, Oudshoorn found that poor housing quality (such as bed bugs, water leakage, etc.), inadequate housing security, and insufficient housing space experienced by Syrian refugees after resettlement to Canada have caused many conflicts [10]. Im and Neff's research on the Bhutan refugee community in the US found that traditional cultural rituals are essential to the identity of new immigrants. Therefore, conflicts caused by immigrants' competition for community public space for traditional religious activities are very common [15]. Kate et al. studied the resettled community in New Zealand and found that the heterogeneity of the population in such communities led to intense conflicts due to competition for community public space resources [13].

Existing studies have conducted many discussions on resettled communities' sustainable development and governance from different perspectives, such as changes in the living environment, deconstruction of social relations, declines in traditional culture, and transformation of lifestyles [3,10,13–15]. However, few studies on the sustainable development of resettlement communities have examined the perspective of community spatial conflicts, whose governance is of great significance to the urban adaptation of landless farmers and the sustainable development of such communities. Thus, we aimed to construct a conceptual framework to systematically explore the generation logic of spatial conflicts in the resettled community for landless peasants and develop targeted governance strategies for the sustainable transformation of such communities.

## 2. Literature Review

The concept of community spatial conflicts derives from the study of social conflict. The discussion of the concept, cause, type, and governance of social conflict has laid a solid theoretical foundation for the research of community spatial conflicts. Some sociologists, such as Coleman, Cosset, and Sanders, put forward the theory of community conflict on this basis [16,17]. However, constrained by the de-spatialization of sociological theories, research on the spatial perspective of community conflict had not received much attention until the 1970s with the "space shift" boom in sociological studies [18]. The space production theory classified by various scholars, for example, Lefebvre, Foucault, Harvey, and Soja, has played an irreplaceable role in increasing research on spatial conflicts [19]. Thereafter, scholars in the fields of geography, ecology, land management, sociology, and economics have discussed the concepts, categories, measures, and causes of spatial con-

flicts. At the city or region level, scholars believe that spatial conflicts arise due to different space uses and their associated external impacts in the process of urban expansion [20]. Most of these spatial conflicts are relevant to the allocation of scarce resources [21]. At the community level, spatial conflict is generally considered a conflict among community members accompanied by a scramble to claim the community's spatial resources [22].

Most of the existing research on spatial conflict in resettled communities has involved the use of questionnaires, semi-structured interviews, and case studies to explore the key factors affecting conflicts. There are three main research perspectives: the perspective of physical space, the perspective of mental space, and the perspective of social space. From the perspective of physical space, scholars generally believe that residents' competition for physical space resources is the direct cause of community spatial conflicts. The essential factors affecting spatial conflicts are housing quality, housing density, housing design, and public facilities [3,10,23,24]. For example, poor housing quality always triggers inter-group conflicts among residents, governments, developers, and other community actors [10]. Meanwhile, the plot ratio in such communities is relatively high, while the space between buildings is very small, resulting in some residents not enjoying enough sunlight and then fighting for spaces for drying with other residents [3]. Moreover, the building form of such a community has changed from an irregular and scattered form of traditional countryside to a relatively vertical centralized cell structure. Changes in spatial patterns lead to the alienation of residents, which is another fundamental cause of spatial conflicts [23].

Second, from the dimension of mental space, factors such as community identity, sense of relative deprivation, sense of resentment, land emotion, and community memory are the psychological roots of community spatial conflicts [5,13,25,26]. For instance, community identity is an element of social cohesion that can reduce community conflict [13]. However, it is not possible to immediately update the peasants' community identity when spaces are transformed between the original rural community and the urban community. This fracture of community identity is one of the factors that prevent peasants from integrating into urban life, and the failure to integrate generates more potential risk of spatial conflicts. The sense of relative deprivation and resentment generated in landless peasants due to demolition and resettlement are the essential emotional roots of spatial conflicts [25]. Furthermore, peasants have a natural affection for the land. Failure to allow the resident to rationally express their emotional connection to the land in the existing living space structure promotes the occurrence of spatial conflicts such as "growing vegetables by destroying the green belt" [5].

Finally, changes in social space elements constitute a potential risk for the generation of community spatial conflicts. Community trust, social network, neighborhood watch, population heterogeneity, community stigmatization, and habitus are important factors [4,27–31]. From the perspective of neighborhood relationships, the resettled community has witnessed a transformation from a traditional society based on blood and geographical ties to an industrial society based on occupation and interests ties. Rapid social changes weaken the relationship network within the community and lower the depth and frequency of communication between neighbors. As a result, it is difficult to rebuild trust relationships in resettled communities, which further threatens the harmony and stability of the community [29]. However, neighborhood watch reduces the probability of conflicts and contributes to resolving conflicts [28].

Although the issue of spatial conflicts in resettled communities has received some attention, some deficiencies exist in previous studies. First, most studies explore the factors influencing spatial conflicts in resettled communities, but the generation mechanism of spatial conflict is still unclear. Second, existing research mostly starts from a single spatial dimension, such as social or physical space, and lacks a systematic analysis framework. Therefore, in this study, we mainly focused on exploring the generation logic of spatial conflicts in resettled communities to provide targeted strategies for the sustainable transformation of such communities. We selected typical resettled communities that have been used to accommodate landless peasants in Hangzhou city, China, as the survey sample.

First, a conceptual model was developed through grounded theory based on first-hand data collected through semi-structured interviews and field observations. Then, we analyzed the inner logic of spatial conflicts generation and applied it in the discussion section of this paper. Ultimately, the results yielded a scientific basis for the transformation of resettled communities, and the findings have implications for similar contexts in other countries and regions.

## 3. Research Methodology

### 3.1. Grounded Theory

Grounded theory is a bottom-up exploratory research method that encourages researchers to explore a field without pre-conceived predictions and construct a new theory from the empirical data of social facts. Given that little work has been done on spatial conflicts in resettled communities for landless peasants, grounded theory is used to construct the conceptual model and explore the generation logic of the spatial conflict. There are three approaches to grounded theory: Classic, Straussian, and Constructivist [32]. In this study, we employed the Straussian grounded theory, which consists of open coding, axial coding, and selective coding, to make the procedures more specific.

### 3.2. Selection of Case

Hangzhou city is the central city of the Yangtze River Delta in China. According to statistics, its urbanization rate was 83.29% in 2020, far above the national average. The construction of resettled communities for landless peasants in Hangzhou started early and developed rapidly in recent years, resulting in a gradual increase in community spatial conflicts. Between July and September 2020, we selected ten typical resettled communities in Hangzhou as the study area (Figure 1). These sample communities have the following characteristics: (1) There are various types. Some communities were built over ten years or more, some were built in five to eight years, and others were built in approximately three years. (2) Population heterogeneity is evident. Community residents include landless peasants resettled from local villages, and landless peasants resettled from other villages, urban residents who buy houses in the community, etc., which bring diverse needs and make the conflicts more complex. (3) There are many news reports related to spatial conflicts in these communities. Therefore, the typicality of the case area was guaranteed.

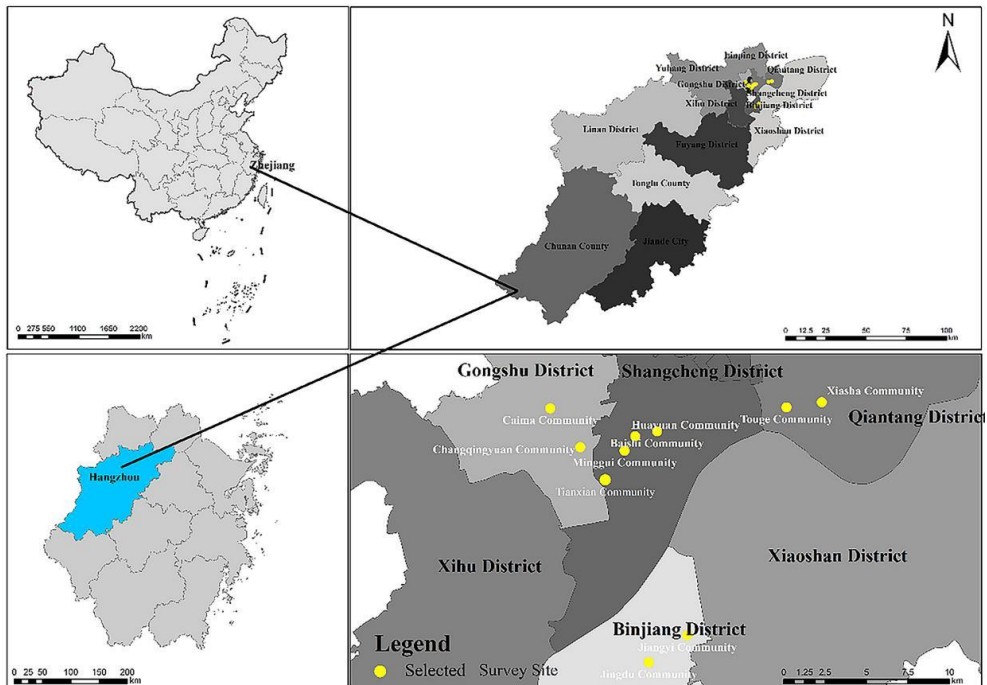

**Figure 1.** Space distribution of selected survey site.

### 3.3. Participants and Interviews

From 2 August to 5 September 2020, we conducted field observations and semi-structured interviews to obtain qualitative data. The grounded theory approach adopts the principle of theoretical sampling, which aims to develop concepts and form a conceptual model; therefore, the sample size does not have to be representative of the whole population [33]. The semi-structured interviews were conducted to obtain the opinions of various stakeholders, and the questions included factors leading to community spatial conflicts, attitudes toward community spatial conflicts, and forms of community spatial conflicts. Twenty-four stakeholders were selected in the following three categories: (1) government officials from district to street level ($n = 4$); (2) community organization staff ($n = 5$), including secretaries of resettled communities and property management company staff; (3) community residents ($n = 15$), including landless peasants, tenants, and urban residents buying houses in the community, with an average interview time of 60 min. With the approval of the respondents, the interviews were recorded, and more than 110,000 words were transcribed.

The interviews' texts were used as original data for the development of initial concepts in the open coding. The development of the initial concept is the basis for constructing the conceptual model [33]. Two-thirds of the interview texts (16 copies) were randomly selected for coding and qualitative analysis, and a conceptual model of the generation logic of spatial conflicts in resettled communities was constructed. The remaining one-third of the interview texts (eight copies) was used for the theoretical saturation test. Interviews and analysis were interrelated, meaning that analysis would be conducted as soon as data were obtained. When the obtained information started to repeat, indicating theory saturation, the interview process was completed. In the process of data analysis, we constantly analyzed, compared, summarized, generalized, and revised the theory until the theory was saturated and formed a new theoretical framework. Two researchers coded at the same time to ensure the credibility of the results. Comparison of results showed that the coded results had high reliability.

### 3.4. Development of Conceptual Model

#### 3.4.1. Open Coding

During the open coding, original data from semi-interviews were read line by line for the purpose of labeling and then coded for a concept. Subsequently, new data were read and compared to the previous data and to choose whether to develop a new concept or revise the former concepts. Based on the principle of constant comparison, we developed 71 initial concepts and 22 categories through the process of labeling, developing concepts, and forming categories. To save space, this paper only shows some concepts and categories and only one sentence representative of each initial concept (Table 1).

**Table 1.** Open coding.

| Serial Number | Category | Original Data (Initial Concept) |
|---|---|---|
| 1 | Quality of private housing | A06 [1]: Water leakage, sound insulation, wall peeling, and other quality problems are common in relocated houses. (Poor housing quality) <br> A18 [1]: The space between buildings is not big enough. (The distance between buildings is small) <br> A03 [1]: Different from the pursuit of quality in commercial housing, the construction of relocated houses is always of a low standard. (Housing construction just meets the minimum standards) |
| 2 | Heterogeneity of community population | A04 [1]: There are people from different places in our community, so there are great difficulties in its integration. (Complex population structure) <br> A06 [1]: There is a great difference in living habits and values between tenants and landless peasants. (There is a big gap in the living habits of different groups) <br> A11 [1]: Tenants have poor maintenance and a low sense of belonging to the community. (Tenants have a low sense of community responsibility) |

| Serial Number | Category | Original Data (Initial Concept) |
|:---:|:---:|:---|
| 3 | Impacts on social psychology | A06 [1]: The most important thing is the accumulated grievances. We farmers have paid so much for the urbanization process in Hangzhou, but our living conditions are also poor. (Sense of resentment towards demolition and resettlement) <br> A05 [1]: The commercial houses' residents look down upon us, and our rich people also look down upon them. (Sense of alienation from urban residents) <br> A12 [1]: It is impossible for us to feel comfortable when a big gap exists between our compensation for land acquisition and others'. (Sense of relative deprivation caused by resettlement policy) <br> A17 [1]: Compared with other communities, the construction conditions of our community are far behind, which is really unfair. (Sense of unfairness caused by the gap in community construction) <br> A24 [1]: We used to have land and houses where we were quite self-sufficient. However, now the whole lifestyle has been changed, and I am not used to it actually. (Sense of loss caused by the decline in life satisfaction) |
| 4 | Trust in community's self-governing organizations | A19 [1]: When we encounter problems, we are accustomed to asking the neighborhood committee for help. (Habitual reliance on neighborhood committee) <br> A09 [1]: If I go to the neighborhood committee to report a problem, tomorrow I will be punished in an underhanded manner. (High cost of safeguarding rights) <br> A09 [1]: Community homeowner committees are useless. They are tied to a pair of trousers with the property management company. (Little truth in the community homeowner committee) <br> A22 [1]: It is two community secretaries that lead to such a big gap between our two communities. A secretary drinks but the other diligently works for the welfare of residents. (Community cadres have charismatic authority) |
| 5 | Awareness of space rights | A15 [1]: I do not approve of burning spiritual money or stacking items in the corridor, because that is our shared space. (Perception of public space rights) <br> A14 [1]: While some residents piled up sundries and made the corridor crowded, considering the relationship with neighbors, others would not stop them. (Claims of public space rights) |
| 6 | Violation of space rights and interests | A01 [1]: Square dancing activity at night will cause noise pollution for some old residents who want to sleep and the young who want to relax. (Square dancing activity violates residents' right to rest) <br> A02 [1]: There was a large-scale conflict in the Tianxian community before, which was caused by the lampblack of the restaurant. (Business operation violates residents' environmental rights) <br> A03 [1]: Residents need to wash clothes and dry them in the sun, leading to frequent contradictions of dripping water from upstairs to downstairs. (Sun-drying violates the lighting rights of other residents) <br> A18 [1]: Go and see our septic tank. It is not merely the unpleasant smell, but accidents may happen. (NIMBY facilities violate residents' environmental rights and safety rights) <br> A23 [1]: The construction of surrounding subway stations caused the relocated house to sink. (The surrounding construction violates the residents' right to live) |

[1] A** represents the original words answered by the ** the respondent.

### 3.4.2. Axial Coding

Axial coding is the process of obtaining the main categories through further refining and classifying all of the categories obtained by open coding. After further clustering and summarizing 22 categories, this paper obtained six main categories: community physical environment, community interaction environment, landless peasants' risk perception, community governance capacity, residents' spatial perception, and space competition behavior. See Table 2 for details of each main category, corresponding initial category, and its connotations.

<div align="center">

**Table 2.** Axial coding.

</div>

| Main Category | Category | Category Connotation |
|---|---|---|
| Community physical environment | Quality of private housing | Quality problems of relocated houses, such as water leakage, sound insulation, and wall peeling |
| | Number of public facilities | The number of infrastructure establishments in the community and support facilities around the community |
| | Function of public space | Whether the function of public space conforms to the planning and whether it meets the needs of residents |
| | Ownership of public space | Whether the ownership of rights of public space is clear or not |
| Community communication environment | Heterogeneity of community population | Differences among community residents in terms of occupation, income, social status, living habits, etc. |
| | Mobility of community population | The ratio of the floating population such as tenants to the residents of the whole community |
| | Harmony of neighborhood relations | The frequency and depth of interaction between community residents |
| | Diversity of cultural life | Diversity of cultural activities of community residents and openness of cultural activities places |
| Landless peasants' risk perception | Basic living security | Whether farmers can meet their basic living needs after losing their means of production |
| | Impacts on social psychology | The negative emotions of the landless peasants caused by changes in community environment, such as the sense of relative deprivation, sense of unfairness |
| | Urban life adaptation | The adaptation of farmers to urban life and sense of identity |
| | Reconstruction of social network | Reconstruction of the relatively stable relationship system |
| Community governance capacity | Trust in community's self-governing organizations | The degree of trust of community residents in neighborhood committees and community homeowner committees |
| | Resident's participation in community autonomy | The frequency and depth of community residents' participation in community public affairs |
| | Service recognition of third-party organizations | Satisfaction of community residents with the services of property management companies and developers |
| | Satisfaction with the grassroots government | Satisfaction of community residents with the governance efficiency of grassroots government |
| Residents' space perception | Awareness of space rights | Be aware of your own space rights, advocate and defend your rights within the scope of the law |
| | Awareness of spatial rules | Respect, awe, and compliance with space rules |
| | Awareness of space ownership | To whom the residents think the space belongs |
| Space competition behavior | Dislocation of space usage | The encroachment of private space on public space caused by individual activities |
| | Fight for public space | Neighborhood committees, property management companies, residents, and other community actors compete for public space usage and interests |
| | Violation of space rights and interests | The use of space by individuals and organizations violates the space rights and interests of others |

### 3.4.3. Selective Coding

The core goal of selective coding is to obtain a conceptual model. The selective coding process can be divided into the following three steps: development of the core category, exploration of the relationship between the core category and other categories, and finally, through the "storyline," the context and conditions of community spatial

conflicts are described. Social combustion theory is an important theory in the field of social physics, which compares social crises and conflicts with combustion phenomena in nature. Therefore, it is applicable for exploring the inner logic of spatial conflicts. The occurrence and development of social crises and conflicts result from the interaction of the three elements: combustion substances, combustion promoters, and ignition temperature [32]. "Combustion substances" are the sources of social disorder; "combustion promoters" are the catalysts of the combustion process; "ignition temperature" is the trigger threshold.

In this paper, the core category is "the spatial conflict in the resettled community for landless peasants." Based on the social combustion theory, the storylines are as follows: in the urbanization process in China, the establishment of the resettled community first realized the transformation of the community's physical environment and community communication environment. The rapid change in the space environment is the source factor; in other words, it is the "combustion substance" of the generation process of spatial conflicts. Thus, landless peasants' risk perception will directly affect their spatial behaviors. Moreover, the transformation of the governance structure of such communities lags behind the transformation of space; therefore, it is difficult for the community governance capacity to match the needs of community spatial governance, which indirectly affects the occurrence of spatial conflicts. Therefore, landless peasants' risk perception and community governance ability have become the two "combustion promoters." Under the catalysis of the combustion promoters, the temperature of the combustion substance gradually increases, and the phenomenon of space competition intensifies. As a contributor to bringing the situation to the "ignition temperature," space competition breaks through the threshold, and then it will lead to community spatial conflicts. Based on the storyline, we constructed a new conceptual model, as shown in Figure 2.

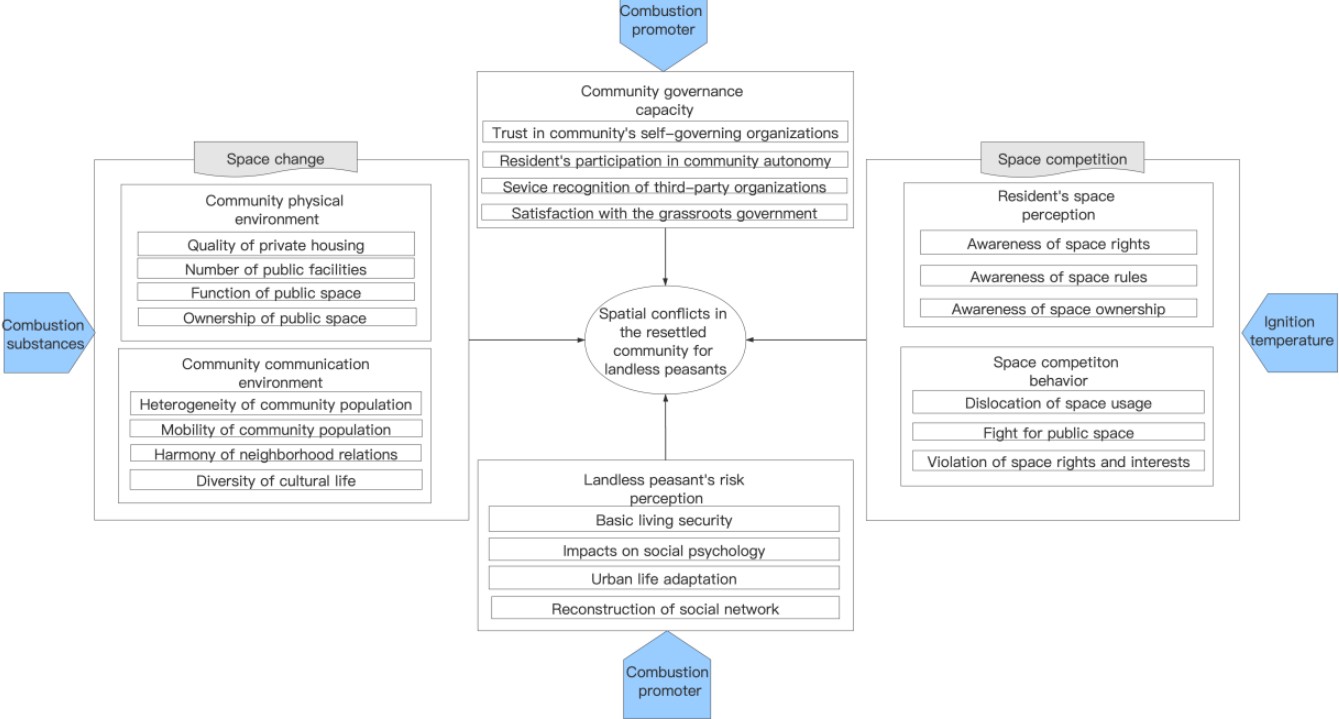

**Figure 2.** Conceptual model of spatial conflict in a demolition and resettlement community.

### 3.4.4. Theoretical Saturation Test

In this study, one-third of the interview texts (eight copies) was used for the theoretical saturation test. The results show that the categories in the model were perfected, which means that the acquired concepts were covered by the existing categories. No new categories were obtained, and no new components were found in each category. Therefore,

this study holds that the conceptual model of spatial conflicts in resettled communities has reached theoretical saturation.

## 4. Generation Logic of Spatial Conflicts in Resettled Community

### 4.1. Space Change: Combustion Substance That Breeds Spatial Conflict

The construction of resettled communities first brings about changes in the private living space of landless peasants. On the one hand, the architectural layout of living space has changed from horizontally dispersed to vertically centralized [27], resulting in an obvious increase in interference among residents. On the other hand, because of the resettlement policy with economic benefits as the main goal [34], the construction of a large number of relocated houses only meets the minimum standards. It has frequently led to quality problems such as water leakage, poor sound insulation, and insufficient lighting, which seriously affect the living environment of landless peasants. This finding corroborates previous studies [10]. Compared with traditional rural communities, there are many problems in the public space of the resettled community, such as greatly reduced scale, lagging design concept, functions failing to meet demand, and fragmented layout. These problems lead to certain behaviors among landless peasants, such as transforming public spaces into private spaces, that can destroy the community's physical environment [6]. This cause-and-effect pattern is verified in this study as the "function of public space" was revealed as an important consideration in how spatial conflicts are generated. We also reported some novel factors affecting spatial conflicts in resettled communities—factors that have not been previously reported. Specifically, compared with modern urban communities, resettled communities for landless peasants generally have problems such as insufficient internal public facilities and inadequate external support facilities. This gap has caused landless peasants to be strongly dissatisfied with the community's living environment. However, owing to the transitional characteristics of such communities, the ownership of public spaces is very vague, reflecting the unclear property rights of underground spaces, parking spaces on a first-come-first-served basis, and the private use of public houses, etc. These are not conducive to the normal use and orderly management of public spaces, causing harmful effects to the physical environment of the community.

Over time, tenants, urban residents buying houses, and other actors continue to move into the community. Consequently, the heterogeneity of the community population continues to increase, gradually forming a situation in which local residents (landless peasants) coexist with the floating population [34]. It is found that because there are notable differences in living habits, community awareness, and values among different groups, it is easy to create contradictions during their communication. It is also found that as housing rental is an important source of income, landless peasants tend to rent out the house, leading to a high rental rate in resettled communities. Therefore, compared with modern urban communities, this community has higher population mobility, which brings great difficulties for community security and building stable neighborhood relations. In the resettlement mode, villagers from the same village move together, maintaining the traditional social ties of acquaintances among the landless peasants. Hence, there is a deep emotional connection among landless peasants.

However, their preference for inter-group communication quickly leads to isolation and opposition between them and the other groups (tenants and urban residents) in the community, negatively impacting harmonious neighborhood relations [8]. This view is also confirmed in this study. In addition, compared with modern urban communities, the diversity of cultural life in this type of community is relatively poor, and the cultural activities of landless peasants tend to be involuted. On the one hand, traditional rural cultural activities such as playing mahjong and having a natter are retained. On the other hand, such gathering activities in the community can easily cause residents to seize the community's public space and other community stakeholders' dissatisfaction with such behavior. This research result is contrary to previous research. Previous research believed that the use of public space contributes to communication among residents [35]. This

difference may be due to the fact that in the resettled communities, such gathering cultural activities occupy a large amount of public space and generate a considerable amount of noise, and the negative externalities toward other residents trigger contradictions and conflicts. Furthermore, many landless peasants are unable to effectively use urban parks and other surrounding support facilities because of their difficulty in adapting to the urban cultural lifestyle. This situation leads to excessive dependence on and the seizure of public facilities within the community.

Demolition and resettlement make the living space of landless peasants undergo cliff-like changes, while it is a gradual process for landless peasants to adapt to urban life [3]. The serious imbalance between time and space has brought about a "negative contribution" to the integration of landless peasants into the new community. Therefore, space change is the root of spatial conflicts in this kind of community, functioning as a "combustion substance". The urbanization practice causes a change in the material form of rural communities. At the same time, the social communication mode of landless peasants also shifted: whereas they were formerly based on blood and geographic relationships, they are now mainly based on career and interest relationships [34]. In conclusion, the changes in the community physical environment and community communication environment are the material and social roots of the spatial conflict, respectively.

### 4.2. Landless Peasants' Risk Perception: Combustion Promoter of Generation of Space Conflict

The resettled community is the basic living space and the main social field of landless peasants, as well as an important place for conflict occurrence [29]. The process of landless peasants' integration into the new environment is accompanied by the reconstruction of their psychological perception [36]. At the same time, serious spatial conflicts occur over time and can be divided into the following four situations. First, landless peasants, who have lost their survival tools, have difficulties in re-employment. Most also rely on the rental income of relocated houses [34]; therefore, their income structure is single. In addition, some landless peasants have encountered difficulties in livelihood, owing to poor management of commercial assets, which come from compensation for demolition. The risk perception of basic living security catalyzes farmers' dissatisfaction with changes in their living environment. They may engage in space competition behaviors, such as rebuilding rental houses and growing vegetables by destroying the green belt to grab more economic income; this research result is consistent with previous research [5]. Second, changing from a rural lifestyle to an urban lifestyle requires a long adaptation period. During this period, the inner psychological perception, such as land emotion, nostalgia for rural living environments, and vague cognition of identity, make it inevitable for landless peasants to continue their rural living habits [5]. It is found that there is insufficient space in the resettled community to meet the needs of these rural living habits. Therefore, conflicts occur because of space use, violation of space rights, and other space competition behaviors. Then, the negative perception of landless peasants on changes in community space is strengthened. Third, after the demolition and resettlement, problems such as the living separation of generations, change of interpersonal communication mode, and weakening of neighborhood trust, make landless peasants face the risk of rebuilding their own social network. Finally, demolition and resettlement not only change their lives but also have different impacts on their social psychology [3]. This study systematically sorts out the negative psychological impact of the demolition and resettlement policies on the landless peasants. These psychological impacts are mainly manifested in the sense of relative deprivation and resentment caused by the resettlement policy, the sense of unfairness caused by the gap in resettlement community construction, the sense of loss caused by the decline in satisfaction with urban life, and a sense of alienation from urban residents. These harm the community communication environment and intensify the competition of landless peasants for community spaces.

It can be seen that the landless peasants' negative perceptions of basic living security, urban life adaptation, social network reconstruction, and social psychology will trigger

their dissatisfaction with the physical and communication environments of the community. Landless peasants' risk perception leads to the exaggeration of these "negative contributions" caused by the change of community space until they become intolerable. Meanwhile, these negative perceptions act as the internal driving force of community space competition, raising the "temperature" to reach the threshold and then leading to spatial conflicts. Therefore, landless peasants' risk perception is the "combustion promoter" for generating spatial conflicts in resettled communities.

### 4.3. Community Governance Capacity: Combustion Promoter to Catalyze Spatial Conflict

Landless peasants in resettled communities have a habitual reliance on administrative power. They used to look for village committees when they encountered problems; thus, they now turn to neighborhood committees for help. In this respect, it is different from other urban communities. Residents in other urban communities will seek help from the property company if they encounter problems [7]. The governance method of neighborhood committees, characterized by undertaking all tasks, makes the governance of the community less effective. Corresponding to this is the extremely low level of participation of residents in creating community autonomy [37]. Two reasons have been explored. On the one hand, the fear of being punished in an underhanded manner makes landless peasants reluctant to participate in community public affairs too often. On the other hand, the desire to participate in community governance power and benefits leads to their dissatisfaction with the establishment of community homeowner committees. These contradictory drives lead to issues such as a homeowner's committee being ineffective and unable to gain credibility. As a transitional community, property management companies, real estate developers, and other third-party organizations are gradually involved in community governance [38]. However, the results of this study reveal that owing to the imperfections in the community governance mechanism, third-party organizations often have problems such as poor service quality and difficult long-term management, which intensify the "negative impact" brought about by the change in community space. For example, a real estate developer's slow process of housing maintenance and negative response will aggravate residents' negative emotions caused by housing quality problems. Previous studies have shown that low-quality property management and low-level property payment rates form a vicious circle [38]. This research further points out that the difficulty of collecting property payment rates will lead to competition between residents and property management companies for the ownership of community spaces, such as advertising and parking spaces. In addition, the grassroots government, as the only stakeholder with law enforcement power, has been absent for a long time in community spatial conflict governance. More precisely, the essential problem is the imperfect interest expression mechanism: residents cannot reflect the real space demand through effective channels, and the grassroots government cannot participate in governance in time during the latent period of conflict generation.

Effective community governance is important to improving the community environment, cultivating residents' benign psychological perceptions, curbing residents' space competition behaviors, and alleviating the opposing emotions among community stakeholders [39]. Therefore, effective community governance is a "flame retardant" for the generation of community spatial conflicts. However, the community governance structure has the characteristics of both rural and urban communities; thus, there are various community governance bodies such as village committees, neighborhood committees, homeowner committees, and property management companies. As urban and rural communities' governance bodies coexist, duplications and overlapping work functions are inevitable and may cause chaos in community governance. Given the capacity of governance of such community governance models, it is difficult for them to adapt to governance needs [40]. As a result, conflicts are not resolved, and community residents are not satisfied with community space governance, all of which indirectly leads to space competition between community stakeholders. In conclusion, community governance capacity becomes the "combustion promoter" to catalyze the generation process of spatial conflicts.

*4.4. Space Competition Fuse of Spatial Conflict*

Space competition behavior is the external manifestation of community space competition and is the most direct cause of community spatial conflicts. This point of view is reported here for the first time. It is concluded that in the resettled community, space competition behaviors include dislocation of space use, fight for public space, and violation of space rights and interests. Dislocation of space use refers to the deviation of residents' space behavior from space function positioning. It mainly includes piling up debris in corridors, illegal construction, raising chickens and ducks, growing vegetables by destroying the green belt, setting up greenhouses for weddings and funerals, and burning spiritual money in corridors. The fight for public space is a competition between different stakeholders in the community for the interests of space [41]. It includes residents' occupation of parking spaces, property management companies' occupation of underground spaces in the community, and neighborhood committees' occupation of supporting houses. It is found that in resettled communities, property management companies operating community public space and grabbing profits is the most common problem, resulting in strong dissatisfaction among community residents. The violation of space rights and interests refers to the negative externalities generated by community stakeholders to other stakeholders in the process of using space [6]. The main forms include square dancing activities that violate residents' right to rest, business operations violating residents' environmental rights, sun drying violating the lighting rights of other residents, and NIMBY facilities violating residents' environmental rights and safety rights. In essence, there is a conflict of space interests between space users and other community stakeholders.

Space perception, involving space right awareness, space ownership awareness, and space rule awareness, is an internal manifestation of community stakeholders' space behavior. The awakening of citizens' rights awareness makes residents defend their own community space rights [42]. In the initial stage of demolition and resettlement, the community is still an acquaintance society with the landless peasant as the main body. The survey results show that because of their relationship, they will not take action against violations of their space rights and interests. Compared with attaching great importance to emotional connections within the group, landless peasants tend to show strong antagonism toward other community stakeholders such as neighborhood committees, property management companies, and tenants [34]. The result shows that the reasons are as follows: First, the landless peasants' awareness of space ownership is very vague. This is reflected not only in landless peasants' cognition of the ownership of community public spaces, such as underground spaces, but also in their cognition of the boundary between public and private spaces. For example, they think that the door, corridor, balcony, and other spaces belong to them, which easily leads to private occupation of public space. Second, the space rules of rural society continue within the community. The awareness of space rules such as "My place, my way" is contrary to the space rules of modern urban communities. With the increase in population heterogeneity in the community, the differences in community residents' space perception are gradually increasing, leading to a continuous increase in space competition perception.

Space competition behavior in the resettled community and residents' space perception are interrelated. This view has not been mentioned in previous related studies. In addition, this research further points out that with the continuous increase of space competition behavior and continuous expansion of space competition perception, the opposition and contradiction of space competition are constantly strengthened. They eventually break the threshold and become the "ignition temperature" that triggers spatial conflict in the community.

**5. Governance Strategies of Spatial Conflict in Resettled Communities**

*5.1. Promote Gradual Transition of Community Space to Create High-Quality Community Environment*

Lots of researchers have pointed out that the change in the lifestyle of landless peasants is gradual, which requires sufficient time to transition [3–5]. Therefore, the construction of community spaces for resettled communities should also adopt a gradual transition mode to reduce the risk of community spatial conflicts from root causes. In terms of the physical environment of the community, the decision-making process for spatial planning of the newly built community must be optimized first. Residents of the community must be given more choices and expression rights to avoid a disconnect between top-down spatial planning and the actual spatial needs of residents [6]. Second, it is necessary to fully guarantee the standards of space construction in such communities and improve the space quality. Finally, the community that has been built should be renovated to optimize the layout of space functions and improve the utilization rate of community space.

In terms of the community communication environment, the advantages of the semi-acquaintance society should be used to promote the reconstruction of a new neighborhood relationship. By increasing the frequency and depth of communication between groups, spatial conflicts can be reduced.

*5.2. Improve Support Mechanism for Landless Peasants to Form Benign Psychological Perception*

Perfecting the social support mechanism and promoting the peasants' adaptation to urban life in terms of livelihood, lifestyle change, psychological integration, and identity are effective ways to eliminate the rejection of sudden changes in community space. There has been extensive research on support mechanisms for landless peasants [4–6,43]. This article advances that the following aspects will help reduce community spatial conflicts by protecting the rights and interests of landless peasants. First, policy support should be provided for the employment of the landless peasants, improving their professional development ability. Second, landless peasants should be guided to form a healthy concept of consumption and investment, and then turn the compensation for relocation into a long-term living guarantee. Furthermore, guidance should be provided that respects the traditional habits of landless peasants while assisting them in gradually forming an urban mindset and lifestyle. Finally, through diversified cultural activities, good communication fields can be built, and loneliness and other negative emotions of the landless peasant can be eliminated.

*5.3. Explore New Community Governance Mechanisms to Enhance Community Governance Capabilities*

Previous studies have shown that establishing a governance mechanism with multiple stakeholders, such as party organizations, grassroots governments, community neighborhood committees, social organizations, and community residents, is the primary way to improve the governance ability of the resettled community [40]. However, the governance mechanism of the resettled communities should match the characteristics of such communities. The improvement of formal mechanisms, such as information sharing mechanism, public participation mechanism, feedback mechanism, social mobilization mechanism, and multi-stakeholder collaborative governance mechanism, will further improve the autonomy of such communities and ultimately improve governance efficiency in the areas of community spatial conflict prevention, resolution, and optimization. In addition, the research results indicate that constructing a flexible integration mechanism can reduce the cost of collaborative governance of community spatial conflicts. For example, the effective governance of community spatial conflicts can be realized by taking advantage of the resources of the semi-acquaintance society in the resettled community, which is precisely the network of relatives and friends among community residents.

*5.4. Cultivate Residents' Awareness of Community Rules and Reduce Space Competition Behaviors*

The key to regulating the space behavior of the resettled community is to cultivate community residents' awareness based on constructing spatial rules that adapt to the transitional characteristics of such communities. On the one hand, it is necessary to clarify the spatial rules of resettled communities and regulate the behavior of community space use. First, community management actors, such as community neighborhood committees and property management companies, need to specify spatial functions, prohibit the misuse of space use, and establish a punishment mechanism for violations. Second, the legal system related to community space rights should be improved. Clarification of community space ownership can reduce public space competition behaviors caused by uncertain ownership [44]. Third, the community should promote the establishment of a compensation mechanism for the violation of space rights by building a platform for negotiation and discussion; ultimately, the distribution of space benefits can be balanced. On the other hand, it is necessary to cultivate the public spirit of community residents and guide them to cultivate an awareness of community rules to improve the platform for community residents to participate in public affairs. At the same time, community activities should be used as carriers to cultivate the public spirit. Through mutual supervision among community residents, residents should be guided to abide by community rules, and, finally, a benign spatial perception can be cultivated.

## 6. Conclusions

The resettled community for landless peasants is a kind of transitional community with features of both rural and urban communities, which brings serious community spatial conflicts. Existing research only mentions the influencing factors of community spatial conflicts, which makes it difficult to provide theoretical guidance for the governance of spatial conflicts and community sustainable transformation. This paper constructs a conceptual model of the generation logic of the spatial conflict in resettled communities. Firstly, ten typical resettled communities in Hangzhou were selected as samples, and 24 interview texts were obtained through field observation and semi-structured interviews. Secondly, using grounded theory to code the interview texts, 71 initial concepts and 22 categories were obtained and further refined into six main categories. The six categories include the community physical environment (e.g., quality of private housing), community communication environment (e.g., heterogeneity of community population), landless peasants' risk perception (e.g., impacts on social psychology), community governance capacity (e.g., trust in community's self-governing organizations), residents' space perception (e.g., awareness of space rights) and space competition behavior (e.g., fight for public space). Finally, social combustion theory was used to construct a logical relationship between the core category and main categories.

Results show that spatial conflict in resettled communities is the result of the interaction of "combustion substance," "combustion promoter," and "ignition temperature". Under the action of the "combustion promoter," the "combustion substance" continuously accumulates in quantity and quality. The "ignition temperature" continues to rise until the threshold is exceeded, and the spatial conflict in the community occurs immediately. Specifically, the community physical and communication environments are the "combustion substance" that causes community spatial conflicts. The diverse experience of the UN-Habitat and the poverty risk and reconstruction models both suggest that resettlement projects bring about numerous social and environmental problems and contradictions [10,24]. This shows that social problems and conflicts are the natural appendages of resettlement. The urbanization process in China has been highly compressed in time and space. It leads to great changes in a community's physical environment and communication environment, even leaving little time for adaptation. This is the root factor of the spatial conflict in the resettled communities. Secondly, landless peasants' risk perception is "combustion promoter," which catalyzes conflicts. Baker and Giddens, the pioneers of risk society

theory, noted that modern society is a risk society [45]. China's landless peasants have been passively thrown into a modern risk society full of uncertainty. Landless peasants' risk perception of, for example, basic living security and urban life, catalyzes the formation of spatial conflict. Thirdly, inefficient community governance capacity also functions as a "combustion promoter," which catalyzes the generation of conflicts. The governance capacity of such a community makes it difficult to match the needs of modern urban communities and boosts the formation of community spatial conflicts. Finally, residents' space perception and space competition behaviors are the fuses that trigger the community spatial conflict, generating the "ignition temperature". However, these space competition behaviors are essentially the necessary actions community residents take to meet their own space needs. It can be understood as a bottom-up space resistance action [46]. Li et al. believe that the less public space occupied and the less impact on other residents, the less likely it is that community space competition will lead to spatial conflicts [6]. This article agrees with this point of view, but we also point out that the resident's space perception affects whether the space competition behavior evolves into community space conflicts or not. The greater the difference in the space perception of community residents or public awareness of the space, the more likely they are to break the threshold and form community spatial conflicts.

However, this research also has some limitations. For example, because the conceptual model of generation logic of spatial conflicts proposed in this article is based on exploratory research, its reliability and validity have not yet been tested by large-sample statistics. In the future, it is still necessary to conceptualize variables involved in the model and develop measurement scales. Large-scale questionnaire surveys should be used to test the relationship among the variables in the model.

Based on the generation logic of community spatial conflicts, the resolution of conflicts should involve measures to reduce combustion substance, increase flame retardant, and reduce ignition temperature for comprehensive prevention and management. Specifically, it includes four aspects: gradual transition of community space, improvement of support mechanism for landless peasants, optimization of community governance mechanism, and cultivation of awareness of community rules. The findings and implications of this study contribute to the existing knowledge, as there are no previous studies examining the generation logic of spatial conflicts in resettled communities. The research findings may be helpful for developing targeted governance strategies for the sustainable transformation of such communities.

**Author Contributions:** Conceptualization, K.X., H.G. and H.B.; Data curation, H.G. and F.Z; Investigation, H.G., F.Z. and J.S.; Methodology, K.X.; Writing—original draft, K.X., H.G. and H.B.; Writing—review & editing, H.B. All authors have read and agreed to the published version of the manuscript.

**Funding:** This research was funded by the Social Science Foundation of Zhejiang Province of China (grant number 19NDQN334YB), National Natural Science Foundation of China (grant number 72004191), MOE (Ministry of Education in China), Project of Science Foundation (grant number 20YJC630174), MOE (Ministry of Education in China), Project of Layout Foundation of Humanities and Social Sciences (grant number 19YJA630001), Soft Science Research Program of Zhejiang Province of China (grant number 2020C25004), and Natural Science Foundation of Zhejiang Province of China (grant number LQ18G030011).

**Institutional Review Board Statement:** Not applicable.

**Informed Consent Statement:** Informed consent was obtained from all subjects involved in the study.

**Data Availability Statement:** The data that support the findings of this study are available from the corresponding author upon reasonable request.

**Acknowledgments:** The authors would like to express their sincere gratitude for the support of ZSSF, NNSFC, MOE, ZSSRP, and ZNSF.

**Conflicts of Interest:** The authors declare no conflict of interest.

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
