# Peer review of "Sustainable Transformation of Resettled Communities for Landless Peasants: Generation Logic of Spatial Conflicts"

_land, doi:10.3390/land10111171_

Round 1

Reviewer 1 Report

The article has been significantly improved. Missing justifications were added after the correction. I have no major objections to the present form of the article. I recommend the paper for publication in Land. 

Reviewer 2 Report

In my opinion the authors properly revised the manuscript.

This manuscript is a resubmission of an earlier submission. The following is a list of the peer review reports and author responses from that submission.

Round 1

Reviewer 1 Report

Dear all,

This working paper deals with interesting issues. Nevertheless, improvements should be consider:

  • the abstract should emphatize the most relevant outcomes of this study
  • the discussion and consluion section should consider more similar studies and researches to foster the debate on this thematic problem and consequently enrich the literature
  • there are no study limitations and prospective research lines - should be added

Regards,

Reviewer 2 Report

Please find the attached documents. 

Reviewer 3 Report

The research is an interesting approach to identifying the mechanism of conflicts in the resettlement areas of landless farmers. This is a universal problem that can apply to migration of people from villages to cities.

The methods used in the context of accepted grounded theory and social combustion theory were well justified. However, the manner in which the survey was conducted raises doubts. The article lacks a detailed description of the preparation and conduct of the interview. When (at what time) was the interview conducted? How were the questions constructed? And finally, how can the responses of specific groups of respondents be correlated with the results obtained? Such data is missing and hence the impression is created that the interview results are subjective and constructed intuitively.

The description of the selected research case should also be supplemented with a map of the research location. Please answer the question whether the distance from and accessibility to the previous place of residence was taken into account as a factor of spatial conflicts? 

Once the necessary clarifications and additions have been made, the article may be published.